# GS-Hider: Hiding Messages into 3D Gaussian Splatting

**Xuanyu Zhang[1,2†], Jiarui Meng[1,2†], Runyi Li[1,2], Zhipei Xu[1,2], Yongbing Zhang[3], Jian Zhang[1,2 ✉]**

[1] School of Electronic and Computer Engineering, Peking University

[2] Guangdong Provincial Key Laboratory of Ultra High Definition Immersive Media Technology, Peking University Shenzhen Graduate School

[3] School of Computer Science and Technology, Harbin Institute of Technology (Shenzhen)

## Abstract

3D Gaussian Splatting (3DGS) has already become the emerging research focus in the fields of 3D scene reconstruction and novel view synthesis. Given that training a 3DGS requires a significant amount of time and computational cost, it is crucial to protect the copyright, integrity, and privacy of such 3D assets. Steganography, as a crucial technique for encrypted transmission and copyright protection, has been extensively studied. However, it still lacks profound exploration targeted at 3DGS. Unlike its predecessor NeRF, 3DGS possesses two distinct features: 1) explicit 3D representation; and 2) real-time rendering speeds. These characteristics result in the 3DGS point cloud files being public and transparent, with each Gaussian point having a clear physical significance. Therefore, ensuring the security and fidelity of the original 3D scene while embedding information into the 3DGS point cloud files is an extremely challenging task. To solve the above-mentioned issue, we first propose a steganography framework for 3DGS, dubbed GS-Hider, which can embed 3D scenes and images into original GS point clouds in an invisible manner and accurately extract the hidden messages. Specifically, we design a coupled secured feature attribute to replace the original 3DGS's spherical harmonics coefficients and then use a scene decoder and a message decoder to disentangle the original RGB scene and the hidden message. Extensive experiments demonstrated that the proposed GS-Hider can effectively conceal multimodal messages without compromising rendering quality and possesses exceptional security, robustness, capacity, and flexibility. Our project is available at: `https://xuanyuzhang21.github.io/project/gshider/`.

## 1 Introduction

As a frontier in computer vision and graphics, 3D scene reconstruction and novel view synthesis are crucial in fields such as movie production, game engines, virtual reality, and autonomous driving. Specifically, thanks to its high fidelity and fast rendering speeds, 3D Gaussian Splatting (3DGS) [21] has become a mainstream approach for 3D rendering. Considering that rendering a 3DGS is extremely costly, protecting the copyright and privacy of 3D assets should be a priority. As a widely studied technique in copyright protection, digital watermarking, and encrypted communication, steganography aims to hide messages like audio, images, and bits into digital content in an invisible manner. In its reveal process, it is only possible for the receivers with pre-defined revealing operations to reconstruct secret information from the container. Therefore, a natural idea arises: *Can we design a steganography method tailored for 3DGS to protect the copyright and privacy of 3D scenes?*

---

[†]Equal contribution. ✉: Corresponding author, zhangjian.sz@pku.edu.cn.

38th Conference on Neural Information Processing Systems (NeurIPS 2024).

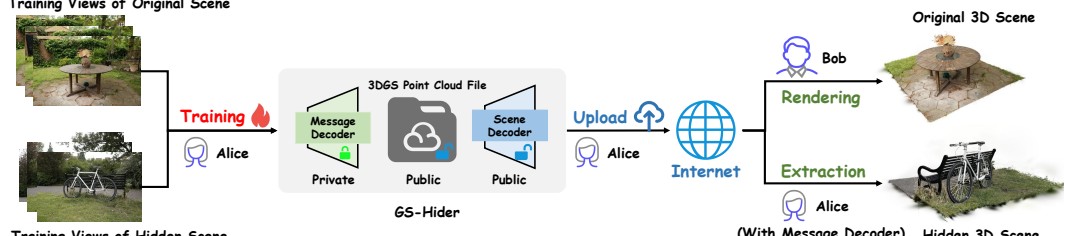

Figure 1: Application scenario of the proposed GS-Hider. The 3DGS trainer (Alice) requires the training views of the original and hidden scenes to train our GS-Hider, comprising a 3DGS point cloud file, a scene and message decoder. Then, Alice will upload the 3DGS point cloud file and the scene decoder online. 3DGS users (Bob) can render the original 3D scene, while only the trainer is authorized to extract the hidden 3D scene, realizing copyright protection or secret communication.

Previous research on 3D steganography has already attracted significant attention. Classical 3D steganography methods [38, 40, 55] often use Fourier and wavelet transforms to embed messages into explicit 3D representations such as meshes and point clouds. During extraction, the receiver must obtain the complete 3D representation. Considering that sometimes only a few views of a 3D scene are publicly available online, the 3D-to-2D watermarking mechanism [60] is designed via a deep encoder-decoder framework, enabling to extract the copyright embedded in the mesh from any 2D perspective. Recently, steganography methods for implicit representations, such as NeRF [35], have emerged. These methods modify the weights of the NeRF [24] or replace the color representation of the NeRF [33], ensuring that each rendered view contains hidden copyright information.

However, the methods mentioned above do not effectively apply to 3DGS due to its unique properties. **First**, since 3DGS is an explicit 3D representation where the attribute of each point has clear physical meanings, we cannot treat it as an implicit representation like NeRF, where the message can be directly and seamlessly embedded into the model weights via optimization [24]. This approach could disrupt the **fidelity** of the rendered RGB views and compromise the integrity of the embedded message. Meanwhile, it is difficult for a single message decoder to faithfully memorize large-capacity hidden messages, such as an entire 3D scene. **Second**, since 3DGS is a real-time renderable 3D representation, users might upload the entire 3DGS point cloud file online and allow others to render it. Thus, the various attributes of the 3DGS are transparent and public, making it difficult to effectively conceal information by simply adding an attribute. Such approaches could lead to significant **security** concerns. **Third**, similar to image steganography [64, 19], we aspire for 3DGS steganography to conceal **versatile** messages (such as 3D scenes, and images) and to explore the capacity limits. Hiding multiple messages allows various users to extract distinct information from the same 3D scene, thereby enhancing the adaptability and engagement of the transmission process. The application scenario of the proposed GS-Hider is presented in Fig. 1.

To solve the above challenges in **security**, **fidelity**, and **functionality**, we propose an effective and flexible steganography framework, dubbed GS-Hider. It aims to embed 3D scenes or images into the original scene, and accurately extract the hidden message via meticulously designed modules. Specifically, it replaces the original 3DGS's spherical harmonics coefficients with a coupled secured feature attribute. Subsequently, a scene decoder and a private message decoder are used to decouple the scene and hidden view from the coupled features in parallel. In a nutshell, the contributions and advantages of our GS-Hider can be summarized as follows.

❑ (1) We present the first attempt to design a 3DGS steganography framework **GS-Hider**. It allows to hiding of messages into a 3D scene in an invisible manner, as well as the exact extraction from the container 3DGS point cloud files. This technology has a broad range of applications in copyright protection of 3D asserts, encrypted communication, and compression of 3DGS.

❑ (2) Our GS-Hider exhibits **robust security and high fidelity**. We ensure the security of GS-Hider by rendering a coupled scene and message feature, supported by a private message decoder. Meanwhile, our method minimally alters the structure of the original 3DGS, while using two parallel decoders that ensure the recovery of original scenes hidden messages do not interfere with each other.

❑ (3) Our GS-Hider exhibits **large capacity and strong versatility**. For the first time, we have realized the ability to hide multiple 3D scenes into a single 3D scene, and at the same time, our GS-Hider can hide a single image into a specific viewpoint of a 3D scene.

❑ (4) We conducted extensive experiments on the 3DGS dataset to demonstrate the security, robustness, fidelity, and flexibility of our method.

## 2 Related Works

### 2.1 3D Representation

The Neural Radiance Fields (NeRF) [35] marked a significant leap in novel view synthesis and multi-view reconstruction. Efforts have focused on improving reconstruction quality [2, 3, 51, 4], enhancing computational efficiency [50, 37, 48, 15, 43, 11], and developing dynamic scene representations [42, 14, 6, 29, 12]. In recent advancements, 3D Gaussian Splatting (3DGS) [21] has emerged as a powerful method for reconstructing and representing 3D scenes using millions of 3D Gaussians. Compared to previous implicit representations such as NeRF, 3DGS offers significant improvements in both training and rendering efficiency. To further enhance the rendering performance of 3DGS, Mip-Splatting [62] achieves high-quality alias-free rendering at arbitrary resolutions by incorporating 2D and 3D filtering. Additionally, Scaffold-GS [30] introduces structured neural anchors, further enhancing the rendering quality of 3DGS from different viewpoints. The superior performance of 3DGS has expanded its applicability to a wide range of fields [54, 46], including SLAM [20, 34], 4D reconstruction [25, 31, 53, 57], and 3D content generation [49, 58].

### 2.2 3D Steganography

Steganography has been evolving over the decades [41, 7]. Thanks to advancements in deep learning, many deep steganography efforts aim to invisibly embed messages into containers and accurately extract them, including 2D images [64, 67, 1, 56, 61], videos [36, 65, 32], audio [26, 27, 8, 44] and generation models [52, 13]. Traditional 3D steganography approaches focus on watermarking explicit 3D representation such as mesh [38, 40, 55] via perturbing these vertices or transforming to the frequency domain. Meanwhile, Yoo et al. [60] aims to extract copyright from each 2D perspective, even when the complete 3D mesh is unavailable. Recently, watermarking implicit neural representations like NeRF have attracted increasing attention [33, 18, 24, 16, 28, 39, 68, 9, 47, 17]. For example, StegaNeRF [24] embedded images or audio into the 3D scene via fine-tuning the NeRF weights. CopyRNeRF [33] built a watermarked color representation and introduced a distortion-resistant rendering strategy to ensure robust message extraction. WaterRF [18] introduced a deferred back-propagation technology with patch loss and resorted to discrete wavelet transform to enhance the fidelity and robustness of NeRF steganography. However, steganography for novel explicit representations 3DGS has not yet been explored.

## 3 Methods

### 3.1 Preliminaries

As shown in Fig. 2, 3DGS is an innovative and state-of-the-art approach in the field of novel view synthesis. Distinguished from implicit representation methods such as NeRF [35], which utilize volume rendering, 3DGS leverages the splatting technique [59] to generate images, achieving remarkable real-time rendering speed. Specifically, 3DGS represents the scene through a set of anisotropic Gaussians, defined with its center position $\boldsymbol{\mu} \in \mathbb{R}^3$, covariance $\boldsymbol{\Sigma} \in \mathbb{R}^{3 \times 3}$ which can be decomposed into scaling factor $\boldsymbol{s} \in \mathbb{R}^3$ and rotation factor $\boldsymbol{q} \in \mathbb{R}^4$, color defined by spherical harmonic (SH) coefficients $\boldsymbol{h} \in \mathbb{R}^{3 \times (k+1)^2}$ (where $k$ represents the order of spherical harmonics), and opacity $\alpha \in \mathbb{R}^1$. Then, the 3D Gaussian can be queried as follows:

$$\mathcal{G}(\mathbf{x}) = e^{-\frac{1}{2}(\mathbf{x}-\boldsymbol{\mu})^\top \boldsymbol{\Sigma}^{-1}(\mathbf{x}-\boldsymbol{\mu})}, \tag{1}$$

where $\mathbf{x}$ represents the position of the query point. Subsequently, an efficient 3D to 2D Gaussian mapping [69] is employed to project the Gaussian onto the image plane:

$$\hat{\boldsymbol{\mu}} = \mathbf{PW}\boldsymbol{\mu}, \quad \hat{\boldsymbol{\Sigma}} = \mathbf{JW}\boldsymbol{\Sigma}\mathbf{W}^\top\mathbf{J}^\top, \tag{2}$$

where $\hat{\boldsymbol{\mu}}$ and $\hat{\boldsymbol{\Sigma}}$ separately represent the 2D mean position and covariance of the projected 3D Gaussian. $\mathbf{P}$, $\mathbf{W}$ and $\mathbf{J}$ denote the projective transformation, viewing transformation, and Jacobian

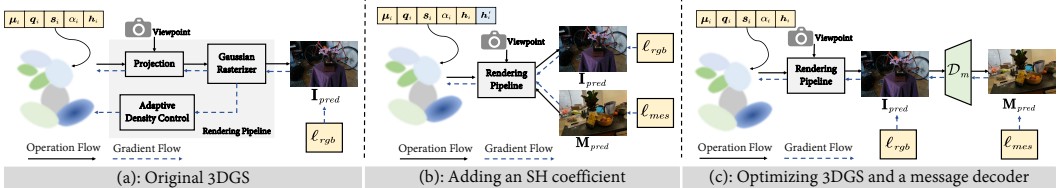

Figure 2: Comparison of original 3DGS and two intuitive approaches of 3DGS steganography, namely adding an SH coefficient, and optimizing 3DGS and a message decoder.

of the affine approximation of $\mathbf{P}$, respectively. The color of the pixel on the image plane, denoted by $\mathbf{p} = (u, v)$, uses a typical neural point-based rendering [22, 23]. Let $\mathbf{C} \in \mathbb{R}^{H \times W \times 3}$ represent the color of the rendered image where $H$ and $W$ represent the height and width of images, the rendering process is outlined as follows:

$$\mathbf{C}[\mathbf{p}] = \sum_{i=1}^{N} \boldsymbol{c}_i \sigma_i \prod_{j=1}^{i-1} (1 - \sigma_j), \quad \sigma_i = \alpha_i \, e^{-\frac{1}{2}(\mathbf{p}-\hat{\boldsymbol{\mu}})^{\top} \hat{\boldsymbol{\Sigma}}^{-1}(\mathbf{p}-\hat{\boldsymbol{\mu}})}, \tag{3}$$

where $N$ represents the number of sample Gaussians that overlap the pixel $\mathbf{p}$. $\boldsymbol{c}_i \in \mathbb{R}^3$ and $\alpha_i \in \mathbb{R}^1$ denote the color calculated from $\boldsymbol{h}_i$ and opacity of the $i$-th Gaussian, respectively.

## 3.2 Task Settings and Some Intuitive Approaches

**Task Settings:** Due to the slow rendering speeds of the implicit representation in NeRF, users often only access a few discrete rendered viewpoints online, rather than obtaining the entire NeRF weights. Consequently, NeRF trainers typically need to embed messages within the model weights and ensure that the same image or bit can be extracted from each rendered 2D viewpoint [24, 33, 18]. However, for 3DGS steganography, due to its real-time rendering capabilities, the trained point cloud files may be directly uploaded online. **Therefore, our task setting is hiding messages during fitting the original 3D scene to create a container 3DGS, and then extracting embedded messages from it.** The difference between ours and the NeRF steganography setup is that: 1) Our extraction requires getting the entire 3DGS point cloud file. 2) Rather than just seeking to extract messages from the rendered 2D view, we focus more on the hiding and extraction in the intrinsic 3DGS point cloud files. Particularly, depending on different purposes, our hidden message can be divided into:

❑ **Encryption Communication**: Hiding 3D scenes in an original 3D scene. We use the original 3D scene to protect secret 3D scenes from malicious theft and extraction by stealers. (Sec. 3.3, Sec. 4.6)

❑ **Copyright Protection**: Hiding an image in a fixed view of the original 3D scene. By comparing a pre-added copyright image with the decoded one, the ownership of the 3DGS is verified. (Sec. 4.6)

In this section, we treat the hidden message as a single 3D scene for clarity. To achieve this task, we first review existing or some potential solutions.

**Original 3DGS**: As shown in Fig. 2 (a), the original 3DGS renders RGB views from learned Gaussian points via a rendering pipeline, including projection, adaptive density control, and Gaussian rasterizer. The learnable attributes of $i$-th 3D Gaussian are represented as $\boldsymbol{\Theta}_i = \{\boldsymbol{\mu}_i, \boldsymbol{q}_i, \boldsymbol{s}_i, \alpha_i, \boldsymbol{h}_i\}$. Here, $\boldsymbol{h}_i$ denotes the SH coefficients which are transformed to $\boldsymbol{c}_i$ and represent the RGB color.

**3DGS+SH**: To embed messages within a 3DGS, as plotted in Fig. 2 (b), an intuitive idea is to introduce another SH coefficient $\boldsymbol{h}_i'$ to fit the hidden 3D scenes, namely learnable parameters $\boldsymbol{\Theta}_i = \{\boldsymbol{\mu}_i, \boldsymbol{q}_i, \boldsymbol{s}_i, \alpha_i, \boldsymbol{h}_i, \boldsymbol{h}_i'\}$. While this approach may achieve moderate fidelity for both the original and the hidden 3D scenes, it compromises security significantly. This is because stealers can easily detect the newly added $\boldsymbol{h}_i'$ in the publicly available point cloud files and simply remove it.

**3DGS+Decoder**: Similar to StegaNeRF [24], another intuitive approach is to add a decoder $\mathcal{D}_m$ to forcibly ensure that the output RGB original views can reveal the corresponding views of the hidden scene. Thus, the learnable parameters are $\boldsymbol{\Theta}_i = \{\boldsymbol{\mu}_i \, \boldsymbol{q}_i, \boldsymbol{s}_i, \alpha_i, \boldsymbol{h}_i\}$ and $\mathcal{D}_m$. However, considering that each Gaussian point has a specific physical significance, jointly optimizing the decoder could potentially compromise the optimization of the 3DGS, decreasing the fidelity of the original scene.

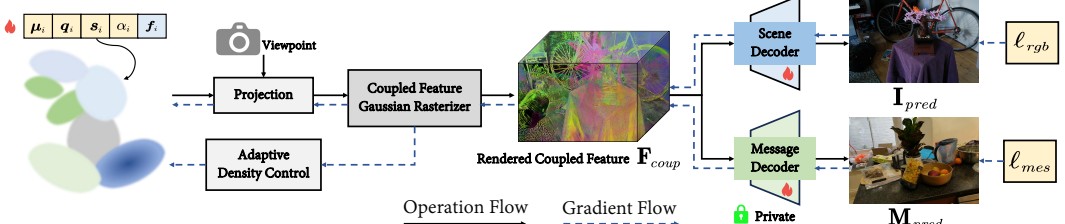

Figure 3: Overview framework of the proposed GS-Hider. It uses a coupled secured feature attribute $\boldsymbol{f}_i$ and the rendering pipeline to fuse hidden and original information, obtaining a rendered coupled feature $\mathbf{F}_{coup}$. Then, the scene and message decoder is adopted to decouple the rendered RGB scenes and hidden messages.

## 3.3 The Proposed 3DGS Steganography Framework GS-Hider

**Motivation:** As depicted in Sec. 3.2, simply adding attributes or treating the 3DGS as a black box and jointly optimizing it with a decoder fails to meet the requirements for 3DGS steganography in **security** and **fidelity**. Thus, we aspire to have a **coupling** and **decoupling** process between the hidden and original information that are no longer presented independently. Inspired by the "Encoder+Decoder" structure in classical image steganography framework [1, 67] and combined with the characteristics of 3DGS, we find that the rendering and training process of 3DGS can be regarded as the "Encoder" for 3D scene fusion, and we can resort to additional deep networks as the "Decoder" to disentangle the original and hidden scene. Furthermore, exploring 3DGS attributes that can securely represent both the original and hidden scenes is challenging. As we have demonstrated in Sec. 3.2, merely adding a SH coefficient is unsafe. Inspired by [66, 25], we find that rendering a high-dimensional feature map, as opposed to merely rendering an RGB image, can contain more information and provide greater confidentiality. Thus, we design a coupled secured feature attribute, a coupled feature rendering pipeline, and two parallel decoders to construct our GS-Hider.

**Defining Coupled Secured Feature Attribute**: Different from $\boldsymbol{h}_i$ in Fig. 2 (a), which has a fixed physical meaning, we define a more flexible attribute $\boldsymbol{f}_i \in \mathbb{R}^M$ to replace $\boldsymbol{h}_i \in \mathbb{R}^{48}$ as shown in Fig. 3, where feature dimension $M$ is arbitrary and adjustable. For $i$-th Gaussian, except for the center position $\boldsymbol{\mu}_i \in \mathbb{R}^3$, scaling factor $\boldsymbol{s}_i \in \mathbb{R}^3$, rotation factor $\boldsymbol{q}_i \in \mathbb{R}^4$ and opacity $\sigma_i \in \mathbb{R}^1$, the coupled secured feature attribute $\boldsymbol{f}_i \in \mathbb{R}^M$ serves to represent the color and textures of both the original 3D scene and the embedded message simultaneously. Defining $\boldsymbol{f}_i$ has two significant benefits. **1)** : It can effectively fuse the original scene and hidden scene via adaptive learning without the need to introduce a separate encoder or additional parameters. **2)** : It is a safe and unified representation, which is impossible for stealers to distinguish which part of the feature represents the original 3D scene and which part represents the message scene.

**Constructing Coupled Feature Rendering Pipeline**: Furthermore, we develop a coupled feature rendering pipeline shown in Fig. 3. Similar to the original 3DGS, we follow the 3D Gaussian initialization via initial set of sparse points from SfM [45] and use the same projection strategy [69] to map the 3D Gaussians to the image plane in a given camera view. Unlike directly rendering an image, we do not need to convert $\boldsymbol{h}_i$ into color component $\boldsymbol{c}_i$. Instead, inspired by Eq. 3, we use a coupled feature Gaussian rasterizer to directly render $\boldsymbol{f}_i \in \mathbb{R}^M$ into the coupled feature $\mathbf{F}_{coup} \in \mathbb{R}^{H \times W \times M}$.

$$\mathbf{F}_{coup}[\mathbf{p}] = \sum_{i=1}^{N} \boldsymbol{f}_i \sigma_i \prod_{j=1}^{i-1}(1 - \sigma_j), \tag{4}$$

where $N$ denotes the number of Gaussians that overlap the pixel $\mathbf{p}=(u, v)$. Specifically, the coupled feature rasterizer uses the tile-based rasterization technique, dividing the screen into $16 \times 16$ tiles, with each thread handling a single pixel. Different from the integration of $\boldsymbol{c}_i \in \mathbb{R}^3$ in Eq. 3, the feature map $\mathbf{F}_{coup}$ is directly rendered in a higher dimension $M$ with greater information capacity. Finally, the coupled feature Gaussian rasterizer enables us to effectively blend the original scene and hidden message via the attribute $\boldsymbol{f}_i$, ensuring the hidden information remains confidential and high-fidelity.

**Disentangling Original and Message Scene:** As plotted in Fig. 3, after obtaining the rendered coupled feature $\mathbf{F}_{coup} \in \mathbb{R}^{H \times W \times M}$, we introduce the scene decoder $\mathcal{D}_s$ to produce the rendered view of the original scene $\mathbf{I}_{pred} \in \mathbb{R}^{H \times W \times 3}$, and design the message decoder $\mathcal{D}_m$ to extract the hidden

Table 1: Comparison of the PSNR(dB) performance of the original and hidden message scenes. We also report the average storage size of 3DGS point cloud files and the weights of decoders (if any). We are not directly comparing with 3DGS. In fact, 3DGS is the **ideal upper limit** of our performance.

| Method | Type | Size (MB) | Bicycle | Flowers | Garden | Stump | Treehill | Room | Counter | Kitchen | Bonsai | Average |
|---|---|---|---|---|---|---|---|---|---|---|---|---|
| 3DGS | Scene | 796.406 | 25.246 | 21.520 | 27.410 | 26.550 | 22.490 | 30.632 | 28.700 | 30.317 | 31.980 | 27.205 |
| 3DGS+SH | Scene | 804.541 | 23.365 | 18.998 | 24.897 | 22.818 | 21.479 | 29.311 | 26.893 | 28.150 | 26.286 | 24.689 |
| | Message | | 23.548 | 25.080 | 28.450 | 24.067 | 20.619 | 22.231 | 20.997 | 22.758 | 21.340 | 23.232 |
| 3DGS+Decoder | Scene | 891.874 | 23.914 | 19.877 | 24.284 | 24.134 | 21.200 | 27.502 | 26.561 | 26.013 | 27.674 | 24.573 |
| | Message | | 20.611 | 20.540 | 25.287 | 19.933 | 19.848 | 21.668 | 20.670 | 22.367 | 20.318 | 21.249 |
| GS-Hider | Scene | 411.356 | 24.018 | 20.109 | 26.753 | 24.573 | 21.503 | 28.865 | 27.445 | 29.447 | 29.643 | 25.817 |
| | Message | | 28.219 | 26.389 | 32.348 | 25.161 | 20.276 | 22.885 | 20.792 | 26.690 | 23.846 | 25.179 |

message scene $\mathbf{M}_{pred} \in \mathbb{R}^{H \times W \times 3}$ as follows.

$$\mathbf{I}_{pred} = \mathcal{D}_s(\mathbf{F}_{coup}), \quad \mathbf{M}_{pred} = \mathcal{D}_m(\mathbf{F}_{coup}). \tag{5}$$

To be concise and ensure real-time rendering, both $\mathcal{D}_s$ and $\mathcal{D}_m$ consist solely of five stacked layers of convolution followed by ReLU activation functions. During deployment, $\mathcal{D}_s$ will be public with the trained 3DGS point cloud file, while $\mathcal{D}_m$ will be kept private as a special protocol, available only to users who are authorized to extract hidden information from the rendered feature $\mathbf{F}_{coup}$. In a nutshell, we use the attribute $\boldsymbol{f}_i$ and rendering pipeline to **couple** the original and hidden information, and employ two decoders for **disentangling** the original and hidden scenes.

### 3.4 Training Details

To train our method, a training set of original scenes $\{\mathbf{I}_{gt}^{(n)}\}_{n=1}^T$ and hidden scenes $\{\mathbf{M}_{gt}^{(n)}\}_{n=1}^T$ that correspond one-to-one in view are required, where $T$ denotes the number of training views. The learnable parameters of our GS-Hider are $\boldsymbol{\Theta}_i = \{\boldsymbol{\mu}_i, \boldsymbol{q}_i, \boldsymbol{s}_i, \alpha_i, \boldsymbol{f}_i\}$, $\mathcal{D}_s$, and $\mathcal{D}_m$. Finally, the training objective of our GS-Hider is defined as:

$$\ell_{rgb} = (1 - \gamma) \cdot \ell_1(\mathbf{I}_{pred}, \mathbf{I}_{gt}) + \gamma \cdot \ell_{SSIM}(\mathbf{I}_{pred}, \mathbf{I}_{gt}), \tag{6}$$

$$\ell_{mes} = (1 - \beta) \cdot \ell_1(\mathbf{M}_{pred}, \mathbf{M}_{gt}) + \beta \cdot \ell_{SSIM}(\mathbf{M}_{pred}, \mathbf{M}_{gt}), \tag{7}$$

where $\gamma$ and $\beta$ respectively denote the balancing weight of $\ell_1$ loss and SSIM loss. Finally, our total loss is $\ell_{total} = \ell_{rgb} + \lambda \ell_{mes}$, where $\lambda$ is used to trade off the optimization between the original scene and the message scene. Similar to 3DGS, during the optimization process, we employ an adaptive density control strategy to facilitate the splitting and merging of Gaussian points.

## 4 Experiments

### 4.1 Experimental Setup

We conduct experiments on 9 original scenes taken from the public Mip-NeRF360 dataset [2]. The correspondence between the hidden and original scene are listed in Tab. 7. $\lambda$ is set to $0.5$ when hiding 3D scenes and set to $0.1$ when hiding a single image. $\beta$ and $\gamma$ in Eq. 6 and Eq. 7 are respectively set to $0.2$. The feature dimension $M$ is set to $16$. We conduct all our experiments on a NVIDIA RTX 4090 Server. Additionally, we modify the original CUDA rasterizer to support the rendering of feature maps of arbitrary dimensions.

Table 2: Rendering time (s) of our proposed GS-Hider.

| Method | Bicycle | Flowers | Garden | Stump | Treehill | Room | Counter | Kitchen | Bonsai | Average |
|---|---|---|---|---|---|---|---|---|---|---|
| GS-Hider | 0.0226 | 0.0145 | 0.0191 | 0.0218 | 0.0218 | 0.0254 | 0.0252 | 0.0255 | 0.0241 | 0.0222 |

### 4.2 Property Study #1: Fidelity

Considering that we are the first to implement 3DGS steganography, we compare the fidelity of our method with the original 3DGS and some intuitive approaches in Sec. 3.2. As shown in Tab. 1, compared to "3DGS+SH" and "3DGS+Decoder", our method achieves 25.817dB and 25.179 dB

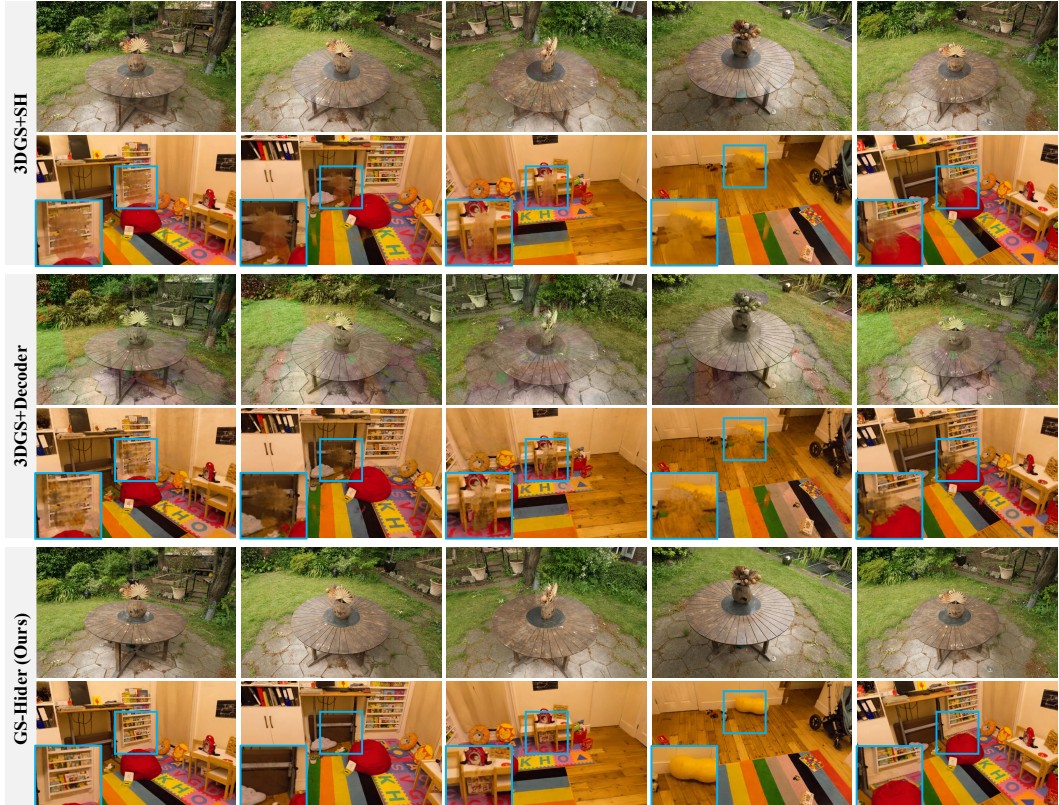

Figure 4: Comparison visualization results of our proposed GS-Hider and other potential methods. The first row of each group: original scene, the second row of each group: hidden scene.

on PSNR for original and hidden scenes, far surpassing other intuitive methods. Compared to the original 3DGS, our method utilizes less storage space (half of the storage size) and incurs only a minimal decrease in rendering performance, while simultaneously possessing the capacity to conceal an entire 3D scene. Note that the performance of the original 3DGS is indeed the upper bound of our method. As plotted in Fig. 4, it is evident that methods like "3DGS+Decoder" or "3DGS+SH" suffer from overlap artifacts between the recovered hidden and original scenes, resulting in limited fidelity and security. However, our method can distinctly reconstruct the two scenes without interference. To analyze on rendering speed of our GS-Hider, We test our GS-Hider on 9 public scenes using a NVIDIA RTX 4090 Server. The resolution of each scene is consistent with the experimental settings of the original 3DGS. The rendering time of the original scene via our GS-Hider is listed in Tab. 2. It can be observed that our method can achieve a rendering speed of 45 fps, which is much **greater than the real-time rendering requirement of** 30 **fps**. This proves the practicality and efficient rendering capabilities of our GS-Hider.

### 4.3  Property Study #2: Security

First, we claim that our coupled feature representation $\mathbf{F}_{coup}$ is secure. This is because it is a high-dimensional, complex, and chaotic feature, and only through our specific message decoder $\mathcal{D}_m$ can the hidden message $\mathbf{M}_{pred}$ be extracted from $\mathbf{F}_{coup}$. Fig. 6 visualizes three random channels of $\mathbf{F}_{coup} \in \mathbb{R}^{H \times W \times M}$ and its corresponding original scene. It can be noticed that the geometric and texture of the feature map are almost consistent with the original scene, and no traces of the hidden information scene can be detected from it. It suggests that the coupled feature field hides messages more

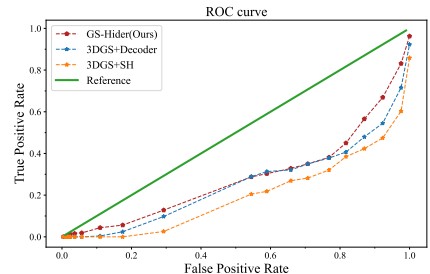

Figure 5: ROC curve of different methods under StegExpose. The closer the curve is to the reference, the method is better in security.

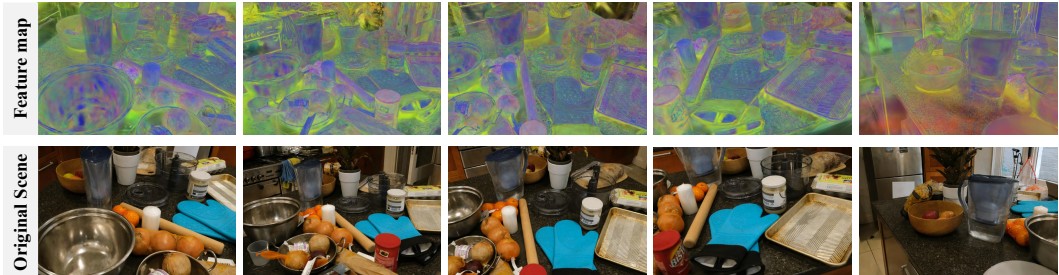

Figure 6: Visualization of the rendered coupled feature map $\mathbf{F}_{coup}$ and the rendering view of the original scene. It can be observed that only the information of the original scene is retained in the feature map, while it is difficult to detect the trace of hidden message scene.

Table 3: Robustness analysis under different pruning methods. $\text{PSNR}_S$, $\text{SSIM}_S$ and LPIPS are used to evaluate the fidelity of the original scene, and $\text{PSNR}_M$ and $\text{SSIM}_M$ are for the hidden scene.

(a) Comparison of sequential pruning ratio.

| Ratio | $\text{PSNR}_S$ | $\text{SSIM}_S$ | LPIPS↓ | $\text{PSNR}_M$ | $\text{SSIM}_M$ |
|---|---|---|---|---|---|
| 5% | 25.804 | 0.783 | 0.245 | 25.179 | 0.780 |
| 10% | 25.804 | 0.783 | 0.245 | 25.179 | 0.780 |
| 15% | 25.804 | 0.783 | 0.245 | 25.179 | 0.780 |
| 25% | 25.771 | 0.782 | 0.332 | 25.167 | 0.780 |

(b) Comparison of random pruning ratio.

| Ratio | $\text{PSNR}_S$ | $\text{SSIM}_S$ | LPIPS↓ | $\text{PSNR}_M$ | $\text{SSIM}_M$ |
|---|---|---|---|---|---|
| 5% | 25.397 | 0.773 | 0.257 | 24.923 | 0.774 |
| 10% | 24.518 | 0.740 | 0.280 | 24.673 | 0.767 |
| 15% | 24.041 | 0.727 | 0.292 | 24.371 | 0.760 |
| 25% | 23.004 | 0.697 | 0.319 | 23.661 | 0.741 |

Table 4: Ablation studies on some key hyper-parameters of the proposed GS-Hider.

(a) Ablation of the balancing weight $\lambda$.

| $\lambda$ | $\text{PSNR}_S$ | $\text{SSIM}_S$ | LPIPS↓ | $\text{PSNR}_M$ | $\text{SSIM}_M$ |
|---|---|---|---|---|---|
| 0.25 | 26.156 | 0.793 | 0.231 | 19.837 | 0.638 |
| 0.5 | 25.817 | 0.783 | 0.246 | 25.179 | 0.780 |
| 1.0 | 24.932 | 0.724 | 0.291 | 28.802 | 0.847 |

(b) Ablation of feature dimension $M$.

| $M$ | $\text{PSNR}_S$ | $\text{SSIM}_S$ | LPIPS↓ | $\text{PSNR}_M$ | $\text{SSIM}_M$ |
|---|---|---|---|---|---|
| 8 | 25.617 | 0.775 | 0.259 | 25.102 | 0.765 |
| 16 | 25.817 | 0.783 | 0.246 | 25.179 | 0.780 |
| 32 | 25.314 | 0.741 | 0.277 | 24.547 | 0.746 |

(c) Ablation of the number of Conv layers.

| Conv | $\text{PSNR}_S$ | $\text{SSIM}_S$ | LPIPS↓ | $\text{PSNR}_M$ | $\text{SSIM}_M$ |
|---|---|---|---|---|---|
| 3 | 25.850 | 0.777 | 0.252 | 24.306 | 0.711 |
| 5 | 25.817 | 0.783 | 0.246 | 25.179 | 0.780 |
| 7 | 25.712 | 0.762 | 0.279 | 25.103 | 0.752 |

to the edges of the object and some imperceptible regions in the artifacts. Furthermore, to verify the security of our GS-Hider, we perform anti-steganography detection via StegExpose [5] on the rendered images of different methods. Note that the detection set is built by mixing rendered images of the original scene and the ground truth with equal proportions. We vary the detection thresholds in a wide range in StegExpose [5] and draw the ROC curve in Fig. 5. The ideal case represents that the detector has a 50% probability of detecting rendered images from an equally mixed detection test, the same as a random guess. Evidently, the security of our GS-Hider exhibits a significant advantage compared to all competitive methods.

## 4.4 Property Study #3: Robustness

To evaluate the robustness of GS-Hider, we have subjected the Gaussians to degradation using both sequential pruning and random pruning methods. Sequential pruning refers to pruning in ascending order of Gaussian's opacity, specifically removing Gaussians with lower opacity first. Random pruning, on the other hand, involves randomly pruning a proportion of Gaussians. Quantitative metrics are shown in Tab. 3. Sequential pruning has minimal impact on the performance of our model, and random pruning also shows minimal effect on the watermarked images. The results indicate that our method effectively withstands the degradation process.

## 4.5 Ablation Studies

Ablation studies on key hyper-parameters of our GS-Hider are presented in Tab. 4. For the parameter $\lambda$, we observe that when $\lambda$=1, the GS-Hider could reconstruct the hidden scene with higher fidelity, making it more suitable for encrypted communication. Conversely, when $\lambda$=0.5, the GS-Hider was better at balancing the recovery of both the hidden and original 3D scenes. Regarding the feature channel $M$ of $\mathbf{F}_{coup}$, we find that the optimal fidelity for both the original and hidden scenes is achieved when $M$ was set to 16. Although the performance with $M$=8 is close to that of $M$=16, lower-dimensional features may result in hidden messages being more easily leaked, compromising security. Regarding the structure of the decoders $\mathcal{D}_m$ and $\mathcal{D}_s$, we test various numbers of "Conv+ReLU" layers. Ultimately, we find that a configuration with five convolution layers allowed the GS-Hider to best balance the reconstruction accuracy of both the original and hidden scenes.

Table 5: PSNR (dB) comparisons between GS-Hider and 3DGS+Decoder on single image hiding.

| Method | Type | Bicycle | Flowers | Garden | Stump | Treehill | Room | Counter | Kitchen | Bonsai | Average |
|--------|------|---------|---------|--------|-------|----------|------|---------|---------|--------|---------|
| 3DGS | Scene | 25.246 | 21.520 | 27.410 | 26.550 | 22.490 | 30.632 | 28.700 | 30.317 | 31.980 | 27.205 |
| 3DGS+Decoder | Scene | 18.320 | 15.224 | 20.901 | 21.884 | 17.435 | 23.878 | 23.322 | 21.174 | 22.481 | 20.513 |
| | Message | 37.210 | 35.564 | 36.228 | 36.548 | 35.844 | 36.924 | 38.833 | 39.261 | 36.157 | 36.952 |
| GS-Hider | Scene | 24.140 | 20.660 | 26.971 | 25.569 | 22.077 | 30.274 | 28.267 | 29.844 | 30.115 | 26.440 |
| (Image) | Message | 39.900 | 43.363 | 39.923 | 39.828 | 39.795 | 39.857 | 42.290 | 47.300 | 50.530 | 42.532 |

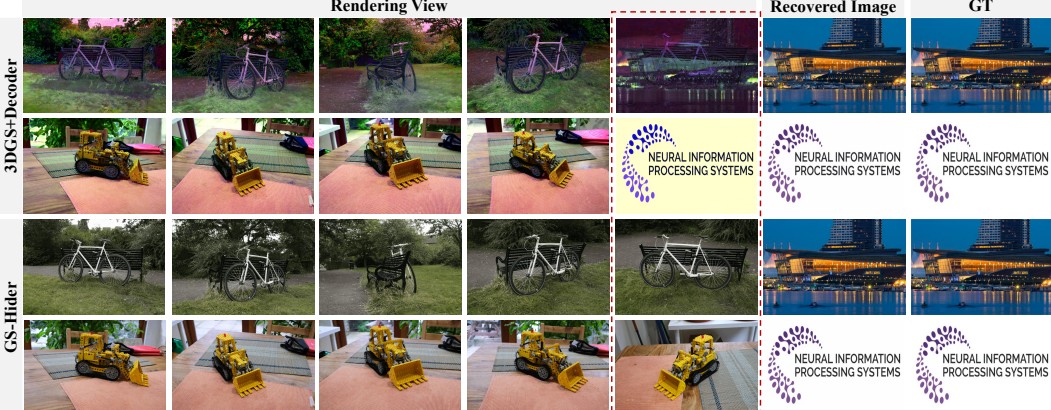

Figure 7: Rendering views and recovered copyright image of our GS-Hider and 3DGS+Decoder. The fifth column of each row represents the rendering view that hides a single image.

### 4.6 Further Applications

**Hiding an image into a single scene:** Embedding an image is a specific case of hiding a 3D scene. For a hidden image $\mathbf{M}_{img} \in \mathbb{R}^{H \times W \times 3}$, we treat it as a fixed viewpoint in the training set $\{\mathbf{M}_{gt}^{(n)}\}_{n=1}^T$. During the fitting of the original scene, we encourage the rendering result of the hidden message at this specific viewpoint to be close to $\mathbf{M}_{img}$ in each iteration, without constraining other views, which makes the $\mathcal{D}_m$ focus on hiding a single image and achieving better fidelity.

To validate the effect of our method for hiding images, we embed an image ("Boat.png") into a **specific viewpoint** of the original 3D scene. Tab. 5 reports the PSNR (dB) of the original 3D scene with the accuracy of the decoded copyright image. It is evident that "3DGS+Decoder" struggles to maintain the fidelity of the original scene when embedding an image. However, our method achieves a copyright image reconstruction performance of 42.532 dB, with only a minor decrease of 0.765 dB in PSNR compared to the original 3DGS. Furthermore, we present the rendered views and recovered hidden copyright image in Fig. 7. Our GS-Hider can accurately reconstruct two different copyright images while causing almost no degradation to the original scene's rendering quality, which proves our method's potential for copyright protection of 3D assets. Note that we show the rendered view of the original scene with an image embedded in the fifth column of each row. Obviously, 3DGS+Decoder is **completely overfitted** to the hidden image (GT) at the specific viewpoint, but our method is immune to the influence of the hidden message.

**Hiding multiple scenes into a single scene:** To embed $L$ hidden scenes into the original 3D scene, we need to modify the last convolution layer of $\mathcal{D}_m$ to $L \times 3$. Then, we jointly optimize $L$ secret scenes and the original scene according to Eq. 7 and 6, which ensures each hidden scene, as well as the original scene, closely approximates the ground truth. To verify the effectiveness of our method for multiple 3D scene hiding, we conceal two groups of hidden scenes into two original scenes. As plotted in Fig. 8, our method can store diverse results of 3D editing [10] within the original 3D scene, reducing the bandwidth load for transmission and presenting different content to different users. Additionally, the GS-Hider is capable of hiding two completely different scenes without interference, maintaining high fidelity.

## 5 Conclusion

We propose a high-fidelity, secure, large-capacity, and versatile 3DGS steganography framework, GS-Hider. By utilizing a coupled secured feature representation with dual-decoder decoding, our method

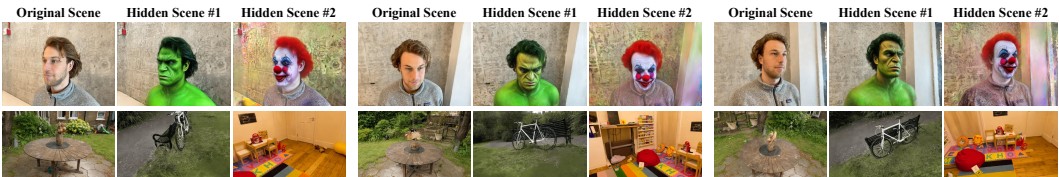

Figure 8: Rendering views of the original 3D scenes, hidden scenes #1, and hidden scenes #2.

can conceal an image, one or multiple 3D scenes in a single 3D scene. To the best of our knowledge, GS-Hider is the first attempt to study 3D Gaussian splatting steganography, which can be applied for encrypted transmission, 3D compression, and copyright protection in 3D asserts. In the future, we will continue to enhance the fidelity and rendering speed of GS-Hider and expand its application scenarios, striving to advance security, transparency, and authenticity in the 3D community.

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

# Appendix

## A Can our GS-Hider decode copyright from arbitrary 2D RGB viewpoint?

Although this paper focuses on hiding and extracting messages from 3DGS point cloud files, our GS-Hider can also extract copyright from any 2D RGB perspective. In some cases, we may not have direct access to the complete 3DGS point cloud file or the rendering pipeline, but can only extract the copyright of the 3DGS from some sparse publicly available 2D views. Thus, similar to the task settings of [24, 33], we design a rendering-resistant traceable watermarking strategy (RTWS) inspired by box-free watermarking approach [63], which allows for the decoding of the copyright from any rendered 2D view by fine-tuning the GS-Hider and a newly added watermark decoder $\mathcal{D}_w$. Specifically, as plotted in Fig. 9, we first use a pre-trained watermark encoder $\mathcal{E}_w$ to embed a shared copyright watermark $\mathbf{W}_{cop}$ into the training set of the original scenes $\{\mathbf{I}_{gt}^{(n)}\}_{n=1}^T$, obtaining watermarked view set $\{\mathbf{I}_{gt}'^{(n)}\}_{n=1}^T$. Subsequently, we mix $\{\mathbf{I}_{gt}'^{(n)}\}_{n=1}^T$ and $\{\mathbf{I}_{gt}^{(n)}\}_{n=1}^T$ in a 50% ratio and use this combined dataset to fine-tune the GS-Hider. To minimize the impact on the original rendering quality, we only alter the coupled feature attribute $\boldsymbol{f}_i$ of the GS-Hider, without changing the positions or shape of each Gaussian point. Finally, we fine-tune the watermark decoder $\mathcal{D}_w$ via the training pairs $\{\mathbf{I}_{pred}, \mathbf{I}_{gt}\}$, constraining it such that when $\mathbf{I}_{pred}$ is input, the decoder outputs $\mathbf{W}_{cop}$, and conversely outputs a black image ($\mathbf{W}_0$) when $\mathbf{I}_{gt}$ is input.

$$\ell_{cop} = ||\mathcal{D}_w(\mathbf{I}_{pred}) - \mathbf{W}_{cop}||_2^2 + ||\mathcal{D}_w(\mathbf{I}_{gt}) - \mathbf{W}_0||_2^2. \tag{8}$$

By adding watermarks and fine-tuning GS-Hider, we alter the training data domain, thereby causing the rendered images $\mathbf{I}_{pred}$ to exhibit domain discrepancies compared to natural images $\mathbf{I}_{gt}$. By fine-tuning the $\mathcal{D}_w$, we enable it to detect this domain gap and decode exact copyrights. The network structure of $\mathcal{E}_w$ and $\mathcal{D}_w$ are similar to [64]. **Note that the proposed RTWS in this section is not only applicable to GS-Hider, but also to original 3DGS, and even other 3D representations such as NeRF.**

To validate the effect of our GS-Hider for extracting copyright from RGB viewpoint, we use the proposed rendering-resistant traceable watermarking strategy (RTWS) to finetune our GS-Hider and a newly added watermark decoder $\mathcal{D}_w$. For simplicity, we choose the pre-trained GS-Hider that has embedded a single image. Tab. 6 reports the PSNR values of the

Table 6: Copyright Extraction Accuracy of Arbitrary 2D Viewpoints.

| Method | PSNR$_S$ | PSNR$_M$ | PSNR$_W$ |
|---|---|---|---|
| GS-Hider (Image) | 26.440 | 42.532 | – |
| GS-Hider (Image) + RTWS | 25.915 | 40.513 | 39.927 |

original scene (PSNR$_S$), hidden message (PSNR$_M$), and the watermarked image extracted from RGB viewpoint (PSNR$_W$). It can be observed that our method is capable of extracting precise copyright watermark images from a 2D view with 39.927dB PSNR via $\mathcal{D}_w$, without sacrificing the fidelity of the hidden message and the original scene.

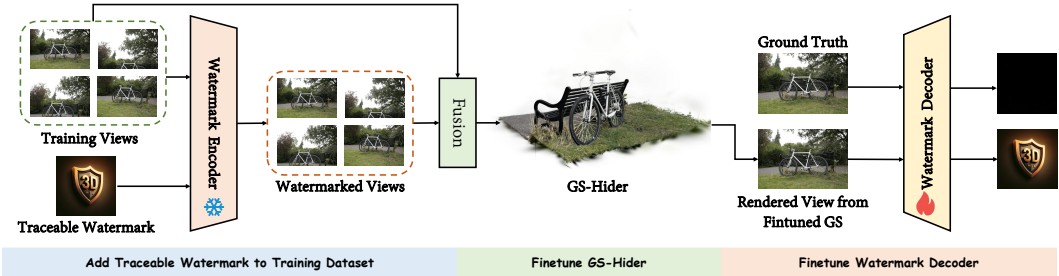

Figure 9: Workflow of the proposed rendering-resistant watermarking strategy.

## B Dataset Construction

To make the training view set of the hidden scene and the original scene correspond to each other, we use the trained 3DGS point cloud files and render them according to the viewpoints in the training set of the original scene to get the training view set of the hidden scene. To ensure that there are as many illegal views as possible in the training views of the hidden scene, we set the correspondence between the hidden and original scenes as listed in Tab. 7.

Table 7: Correspondence between hidden and original scenes. Since for most scenes, "playroom" and "bicycle" have fewer illegal views, they are repeated several times.

| Original Scene | Bicycle | Bonsai | Room | Flowers | Treehill | Garden | Stump | Counter | Kitchen |
|---|---|---|---|---|---|---|---|---|---|
| Hidden Scene | Playroom | Counter | Garden | Playroom | Bicycle | Playroom | Playroom | Bicycle | Bonsai |

## C  Limitations and Future Works

We present the two main limitations of our GS-Hider and provide some potential improvements in the future works. **1) Compromised rendering quality:** Since the feature attribute $f_i$ does not consider view-dependency compared to spherical harmonics, and we need to hide the secret scene while representing the original scene, our rendering quality is somewhat inferior to the original 3DGS. In fact, we inevitably need to make a trade-off between rendering quality and steganography capacity. However, our GS-Hider is a universal framework that can be integrated with the latest 3DGS variants, such as Mip-splatting [62], to enhance rendering performance. Meanwhile, the current scene and message decoder designs are relatively simple. Integrating more efficient neural rendering and decoding designs (such as Scaffold-GS [30]) can also help improve the overall rendering quality of the framework. **2) Decreased rendering speed:** Due to the rasterization of high-dimensional features and network decoding, although we can still achieve real-time rendering, the rendering speed has decreased compared to the original 3DGS. However, we can easily improve rendering speed by pruning Gaussian points, reducing the dimension of feature attributes, and decreasing the number of convolution layers or feature kernels.

## D  Discussions

### D.1  Can the Wrong Decoder Extract the Correct Hidden Scene?

To further verify the security of our GS-Hider, we randomly initialize the message decoder $\mathcal{D}_m$ and use it to decode the rendered coupled feature. As shown in Fig. 10, we find that using the wrong message decoder was completely unable to reconstruct the hidden scene, which further proves that it is difficult for unauthorized users to accurately decode our hidden scene.

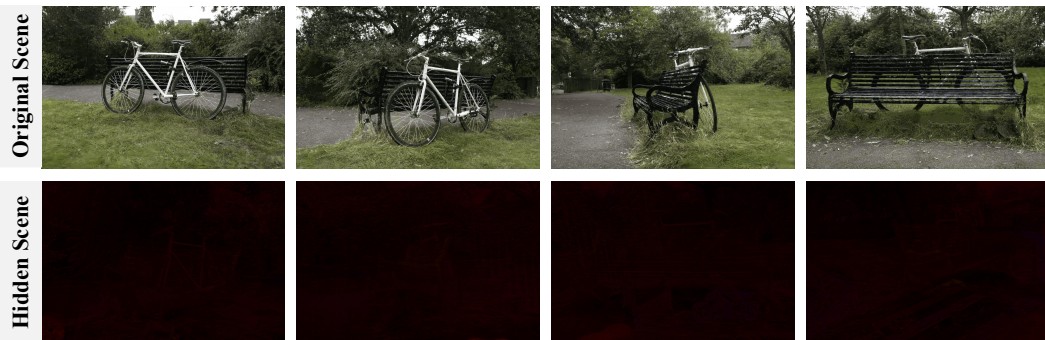

Figure 10: Rendering results produced by the randomly initialized message decoder.

### D.2  Does the Decoder Memorize the Hidden Scene?

To verify whether our GS-Hider simply memorizes the hidden scene via the decoder, we conduct experiments of hiding two hidden scenes, as detailed in Sec. 4.6. This proves that our decoder is not storing or memorizing the secret information. Meanwhile, our message decoder is very lightweight with only **5** convolution layers. It only contains **0.465 M** parameters, which is far from enough to memorize complex 3D scenes. In fact, the geometrical and structural information is mainly embedded in the coupled feature, and the role of the decoder is merely to **extract and decouple the secret information**, not to memorize the scene watermark. To show the role of our decoder, we further input the rendered coupled feature from another scene like 'playroom' to the message decoder that is trained to hide the scene 'bicycle'. The results are presented in Fig. 11. We find that the rendered scene retains most of the geometric structure of the 'playroom' scene, with only some colors resembling

those of the 'bicycle' scene. This indicates that our decoder itself cannot memorize secret information.

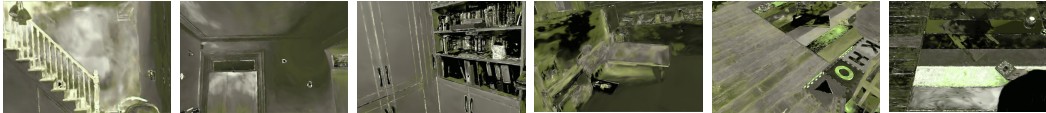

Figure 11: Visualization of the decoded scene when we input coupled features from other scenes ('playroom') to the message decoder of 'bicycle'.

### D.3 Why does the coupled feature attribute work?

First, the hidden scene information is concealed in the spatial high-frequency details of the coupled feature and some visually insensitive areas (such as artifacts, noise, and edges). The invisible hidden information in the coupled feature map will be amplified and decoupled by the message decoder, eventually forming an RGB hidden scene. We visualize the intermediate feature of the message decoder in Fig. 12 to illustrate this process. Second, the secret information is hidden in some redundant feature channels of the coupled feature field $\mathbf{F}_{coup}$. To prove this, we randomly set some channels in $\mathbf{F}_{coup}$ to $\mathbf{0}$, and eventually find that the hidden decoder can not reconstruct the complete secret scene, as presented in Fig. 13. This indicates that multiple feature channels are coupled and interact with each other, collectively storing the hidden information.

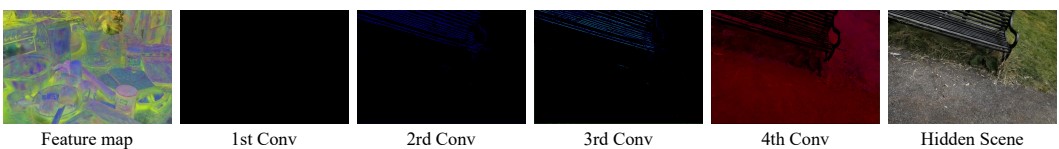

| Feature map | 1st Conv | 2rd Conv | 3rd Conv | 4th Conv | Hidden Scene |

Figure 12: Visualization of intermediate feature maps in the message decoder. We present the 14th-16th channels of the feature map. **Zoom in for best view.**

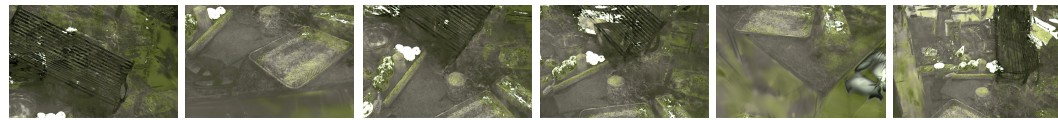

Figure 13: Visualization of the decoded hidden scene when some channels of the coupled feature $\mathbf{F}_{coup}$ is randomly set to $\mathbf{0}$. Obviously, without some channels, the hidden scene cannot be fully decoded correctly.

## E  Additional Quantitative Results

### E.1 Additional Metrics

To verify the performance change of coupled secured feature attributes compared to spherical harmonic coefficients, we do not hide the message but only optimize the 3DGS attributes and scene decoder to fit the original scene. The PSNR, SSIM, and LPIPS of the rendered original scene are listed in Tab. 8. It can be found that without hiding any message, our method only has a PSNR reduction of 0.68 dB compared to the original 3DGS (listed in Tab. 1), which shows that our rendering performance is comparable to 3DGS. Meanwhile, as plotted in Table 1, our storage size is only about half of that of 3DGS. By increasing the feature dimension $M$ and the complexity of the decoder network, our rendering performance can be further improved and approach 3DGS. Finally, due to space limitations, we only put our PSNR results in Tab. 1. We also supplement all metrics of our GS-Hider on single 3D scene hiding in Tab. 9.

### E.2 Comparison with Recent Steganography Method StegaNeRF

To compare with the recent steganography method, we have tried our best to migrate the pipeline and decoding network of StegaNeRF [24] to the 3DGS steganography task. Specifically, we feed the

Table 8: Rendering performance of the proposed GS-Hider without hiding messages.

| Metrics | Bicycle | Flowers | Garden | Stump | Treehill | Room | Counter | Kitchen | Bonsai | Average |
|---|---|---|---|---|---|---|---|---|---|---|
| PSNR | 24.377 | 20.897 | 26.954 | 25.565 | 21.952 | 30.190 | 28.053 | 29.588 | 31.147 | 26.525 |
| SSIM | 0.735 | 0.583 | 0.855 | 0.731 | 0.625 | 0.918 | 0.899 | 0.921 | 0.937 | 0.800 |
| LPIPS | 0.254 | 0.347 | 0.118 | 0.259 | 0.361 | 0.209 | 0.202 | 0.129 | 0.191 | 0.230 |

Table 9: All Metrics of the proposed GS-Hider on the single 3D scene hiding. Note that $PSNR_S$, $SSIM_S$, and $LPIPS_S$ respectively are used to evaluate the fidelity of the original scene, while $PSNR_M$, $SSIM_M$, $LPIPS_M$ are for the fidelity of the hidden message.

| Metrics | Bicycle | Flowers | Garden | Stump | Treehill | Room | Counter | Kitchen | Bonsai | Average |
|---|---|---|---|---|---|---|---|---|---|---|
| $PSNR_S$ | 24.018 | 20.109 | 26.752 | 24.572 | 21.502 | 28.864 | 27.445 | 29.446 | 29.643 | 25.817 |
| $SSIM_S$ | 0.721 | 0.539 | 0.850 | 0.676 | 0.608 | 0.910 | 0.894 | 0.911 | 0.931 | 0.782 |
| $LPIPS_S$ | 0.268 | 0.347 | 0.126 | 0.313 | 0.377 | 0.223 | 0.212 | 0.141 | 0.202 | 0.246 |
| $PSNR_M$ | 28.218 | 26.388 | 32.348 | 25.161 | 20.275 | 22.885 | 20.792 | 26.690 | 23.845 | 25.178 |
| $SSIM_M$ | 0.913 | 0.908 | 0.944 | 0.850 | 0.464 | 0.691 | 0.585 | 0.874 | 0.788 | 0.780 |
| $LPIPS_M$ | 0.210 | 0.245 | 0.137 | 0.287 | 0.487 | 0.350 | 0.497 | 0.208 | 0.328 | 0.306 |

Table 10: Comparison between the proposed GS-Hider and 3DGS+StegaNeRF.

| Methods | $PSNR_S$ | $SSIM_S$ | $LPIPS_S$ | $PSNR_M$ | $SSIM_M$ | $LPIPS_M$ |
|---|---|---|---|---|---|---|
| 3DGS+StegaNeRF | 26.22 | 0.81 | 0.25 | 19.64 | 0.67 | 0.46 |
| GS-Hider | 25.82 | 0.78 | 0.25 | 25.18 | 0.78 | 0.31 |

Table 11: Rendering quality of the extension to Mip-3GDS, namely Mip-GSHider.

| Methods | $PSNR_S$ | $SSIM_S$ | $LPIPS_S$ | $PSNR_M$ | $SSIM_M$ | $LPIPS_M$ |
|---|---|---|---|---|---|---|
| Mip-Splatting | 27.79 | 0.83 | 0.20 | - | - | - |
| Mip-GSHider | 26.25 | 0.79 | 0.24 | 25.26 | 0.76 | 0.34 |

output of 3DGS to the decoding network of StegaNerf and let it approximate the hidden 3D scene. The results are reported on Tab. 10. We find that GS-Hider is much better than 3DGS+StegaNeRF in terms of the reconstruction quality of hidden scenes, achieving 5.54dB improvement. This proves that our GS-Hider can effectively avoid the mutual interference between information hiding and scene rendering.

### E.3 Extension to Mip-3DGS

To verify the generalizability of our framework, we realize a variant of GS-Hider based on Mip-splatting. Specifically, we retain the 3D smooth filter and 2D mip filter from Mip-splatting, only replacing the color attributes with high-dimensional features to fit the GS-Hider framework. Then, we conducted experiments on 3D scenes hiding on the mipnerf-360 dataset. The results are reported in Tab. 10. We also present some visualization results in Fig. 14. This demonstrates that our GS-Hider is a universal steganography framework, not limited to specific 3DGS methods.

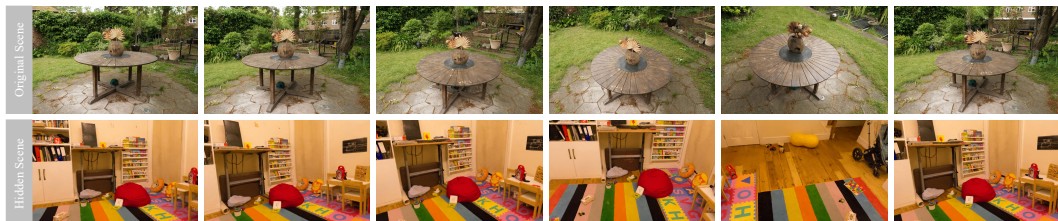

Figure 14: Visualization result of the original and hidden scene rendered by our MIP-GSHider.

## F Additional Visualization Results

We present more visualization results in Fig. 15 and Fig. 16 to demonstrate our effectiveness on 3D scene hiding and single image hiding. Moreover, we constructed an HTML file "./gshider/index.html" in the supplementary material to display some continuous 3D scenes.

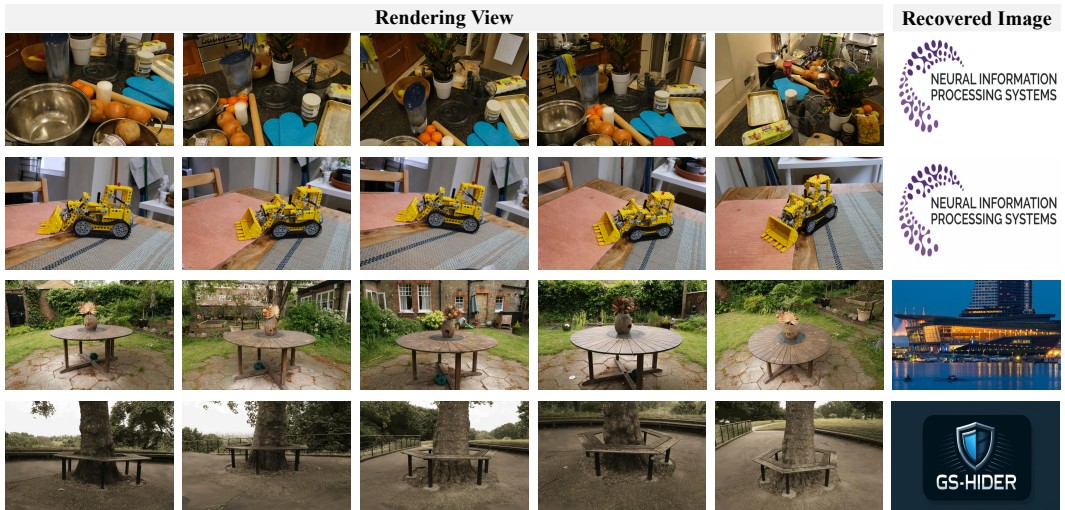

Figure 15: Rendering performance of the original scene and the recovered image produced by our GS-Hider. The fifth column of each row denotes the rendering view that hides a single image.

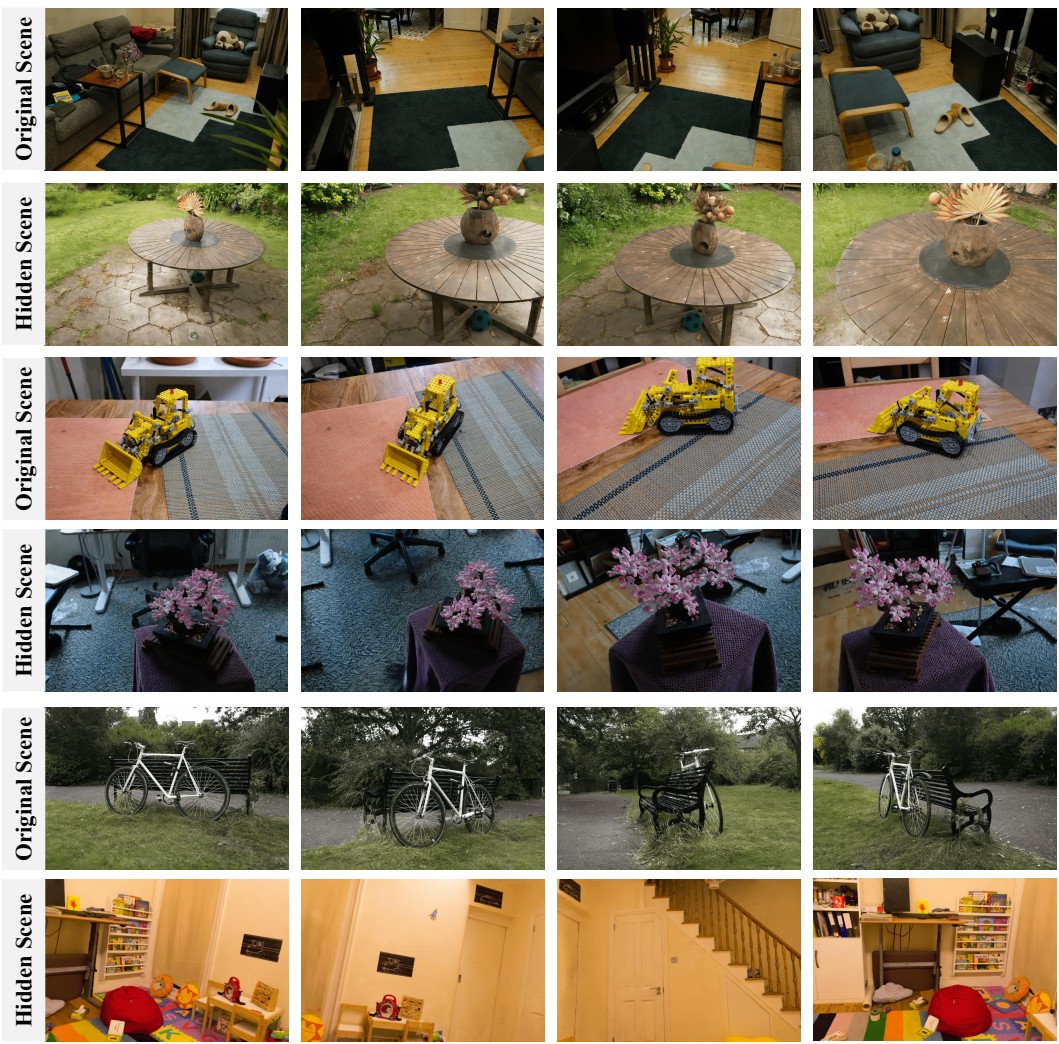

Figure 16: Rendering performance of the proposed GS-Hider on the original and hidden scene.

