# OpenReview forum: "GS-Hider: Hiding Messages into 3D Gaussian Splatting"
_NeurIPS.cc/2024/Conference — NeurIPS 2024 poster_

### Official Review · Reviewer_crVb · 2024-07-03

**Soundness:** 3
**Presentation:** 3
**Contribution:** 3
**Rating:** 6
**Confidence:** 5

**Summary:**

This paper presents GS-Hider, a novel framework for steganography in 3D Gaussian Splatting (3DGS) models. The key innovation is a coupled secured feature attribute that replaces the original spherical harmonics coefficients, allowing the embedding of hidden 3D scenes or images into the original scene without compromising rendering quality. The framework uses a scene decoder and a message decoder to disentangle the original and hidden information. The authors demonstrate the effectiveness of GS-Hider across various experiments, showing its ability to hide multiple 3D scenes or single images while maintaining high fidelity, security, and robustness.

**Strengths:**

Novelty: GS-Hider presents the attempt at steganography for 3D Gaussian Splatting, addressing an important challenge in protecting 3D assets.
Technical innovation: The coupled secured feature attribute and parallel decoder architecture effectively balance security, fidelity, and computational efficiency.
Versatility: The method can hide both 3D scenes and 2D images, demonstrating flexibility for various applications.
Comprehensive evaluation: The authors provide extensive experiments on fidelity, security, robustness, and capacity, comparing against baselines and ablation studies.
Real-time capability: Despite the additional complexity, the method maintains near real-time rendering speeds (45 fps), which is crucial for practical applications.

**Weaknesses:**

Main weakness:
* The main weakness is the additional computational overhead introduced by using the scene decoder. Gaussian splatting significantly simplifies the neural rendering paradigm of NeRF into a point vector + rasterization approach, achieving very fast rendering - this is one of the most valued aspects of GS. However, the introduction of the scene decoder greatly undermines this advantage. This makes such an improvement meaningless. I am confident in this inference because the authors do not seem to report any FPS-related metrics, nor do they compare the FPS increase with the original 3DGS.
* The decoder is trained per scene, rather than being a generalizable decoder, which suggests it may be "memorizing" scene watermarks. This can be inferred from the issues involved in training a decoder for each scene and hiding only a single scene per scene. At the same time, the geometrically inconsistent signals present in the cover scene and hidden scene are well accommodated together, further indicating that the signals are to some extent being "memorized" in the decoder. Therefore, I speculate that the capacity of the decoder is not small. This further corroborates the aforementioned weakness.

Some other weakness:
* While the empirical results are strong, there's a lack of theoretical justification for why the coupled feature representation works so well.
* Comparison to recent work: The paper could benefit from comparing against more recent steganography methods, particularly those designed for other 3D representations that might be adaptable to 3DGS.
* Generalization: The experiments are limited to a single dataset (Mip-NeRF360). It would be valuable to see how the method performs on a wider range of 3D scenes and different types of hidden information.

**Questions:**

Main issues that could be considered in rebuttal：
* what's the added inference time overhead due to using the scene decoder?
* the fps metrics for all experiments, so that we can see the impact on the practical usage. If the rendering speed is affected, the improvement about privacy is not useful.
* try to prove that the hidden scene is not simply memorized by the decoder. maybe trying to memory more than one hidden scenes or curating a general decoder that can be applied to not only one scene can be convincing

Other issues:
please see the weakness

**Limitations:**

The authors provide a brief discussion of limitations in Section B of the appendix, acknowledging issues such as the lack of view-dependency and slightly reduced rendering quality. They also mention future work directions, including enhancing model expressiveness and extending to tasks like tampering detection. However, this discussion could be expanded to more thoroughly address potential failure modes or edge cases of the proposed method.

The broader impacts section (A.2) touches on both positive applications (e.g., copyright protection) and potential risks (sensitive data concerns). However, a more in-depth exploration of potential misuse scenarios and mitigation strategies would strengthen the paper's ethical considerations.

---

> ### Author Rebuttal · Authors · 2024-08-06
>
> Thank you for your constructive comments! If there are any additional comments to be added, please continue the discussion with us.
>
> $\textcolor{red}{\textbf{The supplementary rebuttal PDF file can be found at the bottom of the overall response}}$.
>
> > **Weakness #1, Question #1 and Question #2:  Additional computational overhead.**
>
> - We kindly remind the reviewer that we have already provided the rendering time of each scene in our $\textcolor{red}{\textbf{Table 2}}$ of the main paper and presented the FPS of our GS-Hider in $\textcolor{red}{\textbf{Line 236}}$. It achieves 45 FPS rendering speed, which surpasses the real-time rendering requirement of 30 FPS. Meanwhile, the added inference time of scene decoder (5 Conv layers) is only $\textcolor{red}{\textbf{0.006s}}$, only accounts for $\textcolor{red}{\textbf{0.006/0.0222=27}}$% of the total rendering time.
>
> - The rendering speed of GS-Hider can be **further improved by adjusting some hyperparameters**, such as reducing the number of convolutions in the scene decoder ($N$) and the dimension of feature attributes ($M$), sequentially pruning Gaussian points in ascending order of Gaussian’s opacity. We report the FPS of GS-Hider under different settings and compare it with the original 3DGS on the $\textcolor{red}{\textbf{Table 1}}$ of the rebuttal PDF. The FPS of our lightest GS-Hider ($M=8, N=5$) reaches $\textcolor{red}{\textbf{71.429}}$ and is comparable to the original 3DGS with an acceptable rendering quality.
>
>
>
> > **Weakness #2 and Question #3:  Prove the hidden scene is not memorized by the decoder.**
>
> - We have conducted experiments of hiding two hidden scenes, as detailed in $\textcolor{red}{\textbf{Section 4.6}}$ of the main paper. This proves that our decoder is not storing or memorizing the secret information.
>
> - Our message decoder is very lightweight with only $\textcolor{red}{\textbf{5}}$ convolution layers. It only contains $\textcolor{red}{\textbf{0.465 M}}$ parameters, which is far from enough to memorize complex 3D scenes. In fact, the geometrical and structural information is mainly embedded in the coupled feature, and the role of the decoder is merely to $\textcolor{red}{\textbf{extract and decouple the secret information}}$, not to memorize the scene watermark.
>
> - To show the role of our decoder, we further input the rendered coupled feature from another scene like 'playroom' to the message decoder that is trained to hide the scene 'bicycle'. The results are presented in $\textcolor{red}{\textbf{Figure 4}}$ of the rebuttal PDF.  We find that the rendered scene retains most of the geometric structure of the 'playroom' scene, with only some colors resembling those of the 'bicycle' scene. This indicates that our decoder itself does not have the capability to memorize secret information.
>
>
> > **Weakness #3:  A lack of theoretical justification for why the coupled feature works.**
>
> - The reason of the coupled feature representation works so well is that the feature attribute $\boldsymbol{f}_i$ has sufficient capacity and high flexibility. Combined with our decoder and designed loss function, it can effectively fuse two scenes, and hide secret information in some visually insensitive areas and the redundant feature channels. More analysis can be found in our response to the **Question #1** of reviewer X8jQ.
> - Considering the black-box nature of deep learning, thoroughly analyzing the process of embedding information into high-dimensional features, and providing theoretical proof is an open, challenging, and interesting issue. We appreciate your valuable suggestions and will leave this for future work.
>
>
>
> > **Weakness #4:  Comparison to recent work.**
>
> According to your valuable suggestion, we have tried our best to migrate the pipeline and decoding network of StegaNerf [1] to the 3DGS steganography task. Specifically, we feed the output of 3DGS to the decoding network of StegaNerf and let it approximate the hidden 3D scene. The results are reported on $\textcolor{red}{\textbf{Table 2}}$ of the rebuttal PDF. We find that GS-Hider is much better than 3DGS+StegaNeRF in the reconstruction quality of hidden scenes. If you find other comparison methods suitable for 3DGS steganography, please let us know and we will compare them accordingly.
>
>
>
> > **Weakness #5: Generalization to other datasets and hidden information.**
>
> - **Different dataset:** We have supplemented the experiments on two datasets, namely Tank&Template and Deep Blending, on $\textcolor{red}{\textbf{Table 3}}$ and $\textcolor{red}{\textbf{Figure 5}}$ of the rebuttal PDF. For Tank&Template, we hide the scene 'bicycle' into the two original scenes. For Deep Blending, we insert and hide the 'lego' and 'hotdog' in the Nerf synthetic dataset to the original scenes. It can be found that our GS-Hider can still achieve good results on these two datasets.
>
> - **Different type:** Our GS-Hider can support hiding scenes or a copyright image in our main paper. It can also be applied to hide bits or audio since they can also be treated as a special image. We will realize it in our future work.
>
>
>
> > **Limitations and broader impacts**
>
> - We have reorganized the limitations of our approach and potential improvements in depth in our response to **Weakness #1** of reviewer koxG.
>
> - Our GS-Hider needs to be used in conjunction with an online copyright protection platform to record the exclusive copyrights of different users and scenarios to prevent copyright conflicts. Meanwhile, GS-Hider can be combined with key technology for further protection to prevent users from abusing our models and sensitive data.
>
> > **Reference**
>
> [1] Steganerf: Embedding invisible information within neural radiance fields, in ICCV 2023.

---

> ### Comment · Reviewer_crVb · 2024-08-11
>
> Thank you for your response. It addresses my concerns regarding potential added overhead of inference time. So I raise the rating. I recommend that the revision could include the supplied results in your rebuttal.

---

> > ### Author Response · Authors · 2024-08-11
> > **Thank Reviewer crVb for recognizing our work**
> >
> > Dear Reviewer crVb:
> >
> > We sincerely appreciate your prompt response, valuable suggestions, and recognition of our work. We will include the additional experiments from the rebuttal and provide detailed explanations in the final version to make our paper more rigorous and complete.
> >
> > Best Regards,
> >
> > Authors of #902

---

### Official Review · Reviewer_koxG · 2024-07-04

**Soundness:** 3
**Presentation:** 3
**Contribution:** 3
**Rating:** 6
**Confidence:** 2

**Summary:**

The paper "GS-Hider: Hiding Messages into 3D Gaussian Splatting" proposes a steganography framework for 3D Gaussian Splatting (3DGS). GS-Hider embeds messages into 3D scenes by replacing spherical harmonics coefficients with a secured feature attribute and uses decoders to extract hidden and original scenes without compromising quality. Unlike traditional NeRF methods, 3DGS offers explicit 3D representation and real-time rendering. Experiments show GS-Hider maintains high fidelity, security, and robustness, making it suitable for copyright protection, encrypted communication, and 3D asset compression.

**Strengths:**

1. Innovative Steganography Framework for 3DGS:
GS-Hider is the first framework designed specifically for 3D Gaussian Splatting, allowing for the embedding and extraction of hidden messages within 3D scenes without compromising their fidelity and rendering quality.

2. Robust Security and High Fidelity:
The framework introduces a coupled secured feature attribute and parallel decoders, ensuring the secure and accurate extraction of hidden messages while minimally altering the original 3DGS structure, maintaining high fidelity of the rendered scenes.

**Weaknesses:**

The authors discussed the limitations of their approach, which appear to be relatively minor.

**Questions:**

I am curious about the scenario where an eavesdropper, let's say Eve, downloads Alice's publicly available model, renders it, and then trains her own GS. The purpose of Alice's steganography in Fig. 1 is to verify that the "Table" GS corresponds to her. However, if Eve can train her own "Table" GS, it seems Alice may not be able to protect her "Table" GS effectively. Could you please clarify how this concern is addressed?

**Limitations:**

Please refer to the questions.

---

> ### Author Rebuttal · Authors · 2024-08-06
>
> Thank you for your valuable comments! If there are any additional comments to be added, please continue the discussion with us.
>
> > **Weakness #1:  Minor Limitations.**
>
> Due to limited space, we only discussed the limitations of our method in terms of rendering quality and speed in the supplementary material. We will delve deeper into these two aspects, provide additional insights, and introduce our potential improvements. Our limitations are reorganized as follows.
>
> - **Compromised rendering quality:** Since the feature attribute $\boldsymbol{f}_i$ does not consider view-dependency compared to spherical harmonics, and we need to hide the secret scene while representing the original scene, our rendering quality is somewhat inferior to the original 3DGS. In fact, we inevitably need to make a trade-off between rendering quality and steganography capacity.  However, **our GS-Hider is a universal framework that can be integrated with the latest 3DGS variants**, such as Mip-splatting [1], to enhance rendering performance. Details can be found in the response to **Weakness #2** of reviewer S9BH. Meanwhile, the current designs of the scene and message decoder are relatively simple. Integrating more efficient neural rendering and decoding designs (such as Scaffold-GS [2]) can also help improve the overall rendering quality of the framework.
> - **Decreased rendering speed:**  Due to the rasterization of high-dimensional features and the use of network decoding, although we can still achieve real-time rendering, the rendering speed has decreased compared to the original 3DGS. However, we can easily improve rendering speed by **pruning Gaussian points, reducing the dimension of feature attributes $\boldsymbol{f}_i$, and decreasing the number of convolution layers or feature kernels**. As shown in $\textcolor{red}{\textbf{Table 1}}$ of the rebuttal PDF, these approaches do not significantly impact rendering quality.
>
>
>
> > **Question #1:  How to resist Eve's re-rendering?**
>
> Thank you for presenting such an interesting scenario. If Eve re-renders a GS using Alice's trained model, our copyright protection still holds for the following reasons:
>
> - First, if Eve re-renders a GS, he cannot decode any pre-embedded copyright image or secret scene from it, making Eve's GS unauthorized. Therefore, Eve cannot prove that he owns the copyright of this GS, and the ownership of this GS model still belongs to Alice.
> - Second, our GS-Hider works in conjunction with an online copyright database. When Alice uploads her model, the copyright image is registered in the database. If a similar GS scene is encountered later, the decoded copyright image must be matched with the one in the database. Otherwise, it will be judged as infringement.
> - Third, the cost of re-training a GS is high. Eve would need to spend almost the same computational resources as Alice to steal such a 3D scene, which is difficult for the average thief to achieve.
>
> > **Reference**
>
> [1] Mip-Splatting: Alias-free 3D gaussian splatting, in CVPR 2024.
>
> [2] Scaffold-gs: Structured 3d gaussians for view-adaptive rendering, in CVPR 2024.

---

> > ### Comment · Reviewer_koxG · 2024-08-13
> >
> > Thank you for your response. It has resolved my main concerns. Regarding the adversarial scenario we discussed, I believe revisiting it would be beneficial to ensure preparedness.

---

> > > ### Author Response · Authors · 2024-08-13
> > > **Thank Reviewer koxG for recognizing our work**
> > >
> > > Dear Reviewer koxG:
> > >
> > > We are very grateful for your recognition of our work. We will include your valuable suggestion in the revised version. Your insights have significantly contributed to the improvement of our work.
> > >
> > > Best Regards,
> > >
> > > Authors of #902

---

### Official Review · Reviewer_S9BH · 2024-07-12

**Soundness:** 3
**Presentation:** 3
**Contribution:** 3
**Rating:** 8
**Confidence:** 4

**Summary:**

The paper presents GS-Hider, a novel steganography framework designed for 3D Gaussian Splatting (3DGS). The framework enables the invisible embedding of 3D scenes and images into 3DGS point clouds, ensuring accurate extraction of hidden messages without compromising rendering quality. Extensive experiments demonstrate GS-Hider's effectiveness in concealing multimodal messages while maintaining exceptional security, robustness, capacity, and flexibility.

**Strengths:**

(+) The paper introduces an interesting steganography framework for 3DGS, which is a novel and emerging field in 3D scene reconstruction and rendering. GS-Hider maintains the rendering quality of the original scene while securely embedding hidden messages, addressing the challenges of fidelity and security effectively. The framework has significant potential applications in copyright protection, encrypted communication, and 3D asset compression.

(+) Comprehensive experiments are conducted to validate the performance, security, robustness, and flexibility of GS-Hider. The experiment results demonstrate robustness against various forms of degradation and support hiding multiple 3D scenes or images, showcasing its versatility.  Further Applications effectively explain the results of the proposed method when dealing with other scenarios.

**Weaknesses:**

(-) The implementation of GS-Hider involves some techniques and may require substantial computational resources, limiting its accessibility and usability for some ordinary users without deep learning backgrounds.
(-) The comparison with existing methods is somewhat limited, as it primarily focuses on a specific type of 3DGS. It overlooks a broader range of state-of-the-art techniques in 3DGS. It may be beneficial for the author to consider implementing their methods in other 3DGS variants like [1] to highlight the advantages of the proposed method.

[1] Mip-Splatting: Alias-free 3D Gaussian Splatting.

**Questions:**

Please see the weakness.

**Limitations:**

The authors have addressed the limitations in the paper.

---

> ### Author Rebuttal · Authors · 2024-08-06
>
> Thank you for your constructive comments! We hope that our responses can address your concerns. If there are still aspects that need further clarification, please feel free to continue the discussion with us!
>
> > **Weakness #1: Limited accessibility and usability.**
>
> - Our method is actually very simple, efficient, and user-friendly. It does not require substantial computational resources than the original 3DGS. In fact, our storage space is even smaller than the original 3DGS because we use a more compact feature representation.
> - Additionally, our method can be optimized end-to-end in a $\textcolor{red}{\textbf{single GTX 3090 Ti Server}}$, with training and decoding easily encapsulated into an interface. **In the training phase**, users only need to input the original scene and the hidden scene or a copyright image to render a secure and private 3DGS for publishing or sharing. **In the verification phase**, users only need to use a private message decoder to extract encrypted information from the Gaussian point cloud. This makes it accessible for users without a background in deep learning to use it effortlessly.
>
>
> > **Weakness #2: Extension to other variants of 3DGS.**
>
> - According to your valuable suggestions, we realize a variant of GS-Hider based on Mip-splatting. Specifically, we retain the 3D smooth filter and 2D mip filter from Mip-splatting, only replacing the color attributes to high-dimensional features $\boldsymbol{f}_i$ to fit the GS-Hider framework. Then, we conducted experiments on 3D scene hiding on the mipnerf-360 dataset. The results are reported as follows. We also present some visualization results in $\textcolor{red}{\textbf{Figure 3}}$ of the rebuttal PDF file. This demonstrates that our GS-Hider is a universal steganography framework, not limited to specific 3DGS methods.
>
>   | Method             | $\text{PSNR}_S$ | $\text{SSIM}_S$ | $\text{LPIPS}_S$ | $\text{PSNR}_M$ | $\text{SSIM}_M$ | $\text{LPIPS}_M$ |
>   | ------------------ | ------ | ------ | ------- | ------ | ------ | ------- |
>   | Mip-Splatting      | 27.79  | 0.83   | 0.20    | -      | -      | -       |
>   | Mip-GSHider (Ours) | 26.25  | 0.79   | 0.24    | 25.26  | 0.76   | 0.34    |

---

> > ### Comment · Reviewer_S9BH · 2024-08-10
> > **Thanks**
> >
> > Thanks for your detailed response, which has addressed my concerns. After reading the rebuttal and other reviews, I believe this work has practical applications in copyright protection of 3D assets. Meanwhile, the author implemented a variant of GS-Hider based on mip-splatting in the rebuttal, verifying that the proposed method is a general framework. Therefore, considering the novelty of this paper in 3D steganography, good rendering quality and real-time rendering speed, I decide to raise my score to 8. However, please also remember to address my concerns in your final version, especially those parts that you have promised.

---

> > > ### Author Response · Authors · 2024-08-10
> > > **Thank Reviewer S9BH for recognizing our work**
> > >
> > > Dear Reviewer S9BH:
> > >
> > > Thank you for your prompt response and for acknowledging our work. We sincerely appreciate your valuable suggestions and assure you that we will include the additional experiments and explanation from the rebuttal in the final version.
> > >
> > > Best regards,
> > >
> > > Authors of #902

---

### Official Review · Reviewer_X8jQ · 2024-07-12

**Soundness:** 3
**Presentation:** 3
**Contribution:** 3
**Rating:** 6
**Confidence:** 2

**Summary:**

The paper introduces GS-Hider, a novel steganography framework for 3D Gaussian Splatting (3DGS). Protecting the security and fidelity of 3D assets while embedding information into transparent 3DGS point clouds is challenging, and the method addresses this by invisibly embedding 3D scenes and images into original GS point clouds. It employs a coupled secured feature attribute, scene decoder, and message decoder. Extensive experiments demonstrate its effectiveness in concealing multimodal messages without compromising rendering quality a lot.

**Strengths:**

1) The first work to perform steganography for the 3D gaussian splatting, it might inspire more research into this direction.
2) The method works well and exhibits robustness and high capacity shown by the empirical results.

**Weaknesses:**

1) The rendering quality and speed are compromised a bit, but not much.

**Questions:**

1) It is interesting to understand where and how exactly your method hides the secret information. As you show in Figure 6, at first glance, it seems that the rendered coupled feature map only contains information about the original scene. Does it mean that the hidden message is encoded as (spatial) high-frequency details in the feature channels? Or maybe it learnt to hide information in last bits just like the least significant bit method? Are there any experiments that you conducted to investigate this more?

**Limitations:**

Limitations are adequately addressed.

---

> ### Author Rebuttal · Authors · 2024-08-06
>
> Thank you for your constructive comments! We hope that our response will address all of your concerns. All discussions and supplementary analyses will be included in our revised version. If there are any additional comments to be added, please continue the discussion with us.
>
> $\textcolor{red}{\textbf{The supplementary rebuttal PDF file can be found at the bottom of the overall response}}$.
>
> > **Weakness #1: Compromised rendering quality and speed.**
>
> - First, our rendering quality and speed are still $\textcolor{red}{\textbf{acceptable}}$.
>   - In the case of hiding a 3D scene, our rendering quality only drops a little bit (about 1dB) and is comparable to the original 3DGS and other 3D rendering methods (such as Instant NGP).
>   - Meanwhile, our rendering speed can reach 45 FPS, $\textcolor{red}{\textbf{which far exceeds the real-time rendering requirement of 30 FPS}}$.
> - Second, our method has $\textcolor{red}{\textbf{unique advantages}}$ compared with the original 3DGS.
>   - As shown in Table 1 of the main paper, our GS-Hider significantly reduces the storage space by $\textcolor{red}{\textbf{385.05MB}}$ compared to the original 3DGS.
>   - Our GS-Hider has better privacy and can hide a 3D scene or a copyright image with high quality, which is suitable for tasks such as encrypted transmission and copyright protection.
>
> - Third, the rendering speed of our GS-Hider can be $\textcolor{red}{\textbf{easily improved}}$.
>   - We can apply GS compression methods to reduce the number of Gaussian points and improve rendering speed. As shown in Table 2 of the main paper, sequentially pruning Gaussian points by 25% hardly affects the rendering quality and can bring a **20%** improvement in rendering speed.
>   - We can improve rendering speed by reducing the number of convolutions in the scene decoder or using a more efficient and lightweight network design. More FPS results of GS-Hider under different settings can be found on $\textcolor{red}{\textbf{Table 1}}$ of the rebuttal PDF.
>
> > **Question #1: Where and how GS-Hider hides the secret information?**
>
> - First, the hidden scene information is concealed in the **spatial high-frequency details of the coupled feature and some visually insensitive areas (such as artifacts, noise, and edges)**. The invisible hidden information in the coupled feature map will be amplified and decoupled by the message decoder, eventually forming an RGB hidden scene. We visualize the intermediate feature of the message decoder in the $\textcolor{red}{\textbf{Figure 1}}$ of the rebuttal PDF to illustrate this process.
>
> - Second, the secret information is hidden in some **redundant feature channels** of the coupled feature field $\mathbf{F}\_{coup}$. To prove this, we randomly set some channels in $\mathbf{F}\_{coup}$ to $\mathbf{0}$, and eventually find that the hidden decoder can not reconstruct the complete secret scene. The results are presented in $\textcolor{red}{\textbf{Figure 2}}$ of the rebuttal PDF. This indicates that multiple feature channels are coupled and interact with each other, collectively storing the hidden information.

---

> > ### Comment · Reviewer_X8jQ · 2024-08-12
> >
> > I thank the authors for their response. I will keep my score.

---

> > > ### Author Response · Authors · 2024-08-12
> > > **Thank Reviewer X8jQ for recognizing our work**
> > >
> > > Dear Reviewer X8jQ:
> > >
> > > Thank you for your response and for recognizing our work. We will include the content from the rebuttal in the final version.
> > >
> > > Best Regards,
> > >
> > > Authors of #902

---

### Author Rebuttal · Authors · 2024-08-06

We sincerely appreciate all the constructive comments from the reviewers! Below is our brief overall response.

> **First, we are very honored to receive recognition from all the reviewers for various aspects of our work.**

- All reviewers have acknowledged the **soundness, presentation, contribution, and effectiveness** of our GS-Hider.
- All reviewers have recognized the **innovation** of GS-Hider in 3DGS steganography and consider it an interesting and important work.

> **Second, we would like to emphasize the value and contribution of our work.**

- GS-Hider is **the first attempt** at 3DGS steganography, which can be applied to the encrypted transmission and copyright protection of 3D assets,  serving as inspiration for the future development of 3DGS steganography.
- GS-Hider presents **unique advantages** in security, privacy, robustness, and versatility, while having acceptable rendering quality and real-time rendering speed.

> **Third, we have tried our best to address all of the concerns raised by reviewers and added detailed analysis.**

- Regarding concerns on rendering speed raised by reviewer X8jQ and crVb, we list the added inference time by the scene decoder, present the comparison between GS-hider under different settings and original 3DGS, and give **potential improvement** methods.
- We realize **a variant of GS-Hider based on Mip-splatting**, **supplement the experiments on other datasets**, and **compare with more methods** to make our paper more comprehensive and rigorous.
- We further analyze why the coupled feature representation is effective, explore the limitations of our approach, and clarify some application scenarios.

We sincerely thank all the reviewers for their suggestions to improve our paper and kindly request the reviewers to thoroughly consider the value and contribution of our work. The additional experiments and analyses will be added to the final version of the paper. The detailed rebuttals for each reviewer can be found below.

Additionally, we have attached a $\textcolor{red}{\textbf{PDF}}$ file containing some figures and tables for the reviewers' reference.

---

### Author Response · Authors · 2024-08-10
**Looking forward to discussions with reviewers**

Dear Reviewers:

We appreciate the time you dedicated to reviewing our work and your recognition of our work. Regarding the concerns you raised, we have provided explanations in our responses.

We would like to ensure that your concerns have been adequately addressed. If there are any aspects of our work that remain unclear to you, please don't hesitate to let us know.

Thank you for your dedication!

Best regards,

Authors of #902

---

### Comment · Area_Chair_tBjJ · 2024-08-11

Dear Reviewers,

Authors have provided their rebuttal. Could you please take a look to see whether your concerns are addressed or not? Thanks!

---

### Decision · Program_Chairs · 2024-09-25

**Decision:**

Accept (poster)

**Comment:**

Before rebuttal most of the reviewers have concerns on the computational cost, etc. After rebuttal, all reviewers agreed that authors resolved their concern and the final rating of this paper is 1 strong accept and 3 weak accept. So AC decision to accept this paper.